# Psychometric Evaluation of Three Versions of the UCLA Loneliness Scale (Full, Eight-Item, and Three-Item Versions) among Sexual Minority Men in Taiwan

**DOI:** 10.3390/ijerph19138095

**Published:** 2022-07-01

**Authors:** Chung-Ying Lin, Ching-Shu Tsai, Chia-Wei Fan, Mark D. Griffiths, Chih-Cheng Chang, Cheng-Fang Yen, Amir H. Pakpour

**Affiliations:** 1Institute of Allied Health Sciences, College of Medicine, National Cheng Kung University, Tainan 70101, Taiwan; cylin36933@gs.ncku.edu.tw; 2Biostatistics Consulting Center, National Cheng Kung University Hospital, College of Medicine, National Cheng Kung University, Tainan 70101, Taiwan; 3Department of Public Health, College of Medicine, National Cheng Kung University, Tainan 70101, Taiwan; 4Department of Occupational Therapy, College of Medicine, National Cheng Kung University, Tainan 70101, Taiwan; 5Department of Child and Adolescent Psychiatry, Chang Gung Memorial Hospital, Kaohsiung Medical Center, Kaohsiung 83301, Taiwan; jingshu@cgmh.org.tw; 6School of Medicine, Chang Gung University, Taoyuan 33302, Taiwan; 7Department of Occupational Therapy, AdventHealth University, Orlando, FL 32803, USA; chia-wei.fan@ahu.edu; 8International Gaming Research Unit, Psychology Department, Nottingham Trent University, Nottingham NG1 4FQ, UK; mark.griffiths@ntu.ac.uk; 9Department of Psychiatry, Chi Mei Medical Center, Tainan 70246, Taiwan; 10Department of Health Psychology, Chang Jung Christian University, Tainan 71101, Taiwan; 11Department of Psychiatry, School of Medicine College of Medicine, Kaohsiung Medical University, Kaohsiung 80708, Taiwan; 12Department of Psychiatry, Kaohsiung Medical University Hospital, Kaohsiung 80756, Taiwan; 13College of Professional Studies, National Pingtung University of Science and Technology, Pingtung 91201, Taiwan; 14Department of Nursing, School of Health and Welfare, Jönköping University, SE-551 11 Jönköping, Sweden; amir.pakpour@ju.se

**Keywords:** bisexual, gay, homosexual, loneliness, UCLA Loneliness Scale, psychometric properties, psychological well-being

## Abstract

The UCLA Loneliness Scale, with different short versions, is widely used to assess levels of loneliness. However, whether the scale is valid in assessing loneliness among sexual-minority men is unknown. Additionally, it is unclear whether the 8-item and 3-item short versions are comparable to the full 20-item version. The present study compared the validity of the three versions of the UCLA Loneliness Scale (i.e., 20-item, 8-item, and 3-item versions) among gay and bisexual men in Taiwan. The participants comprised 400 gay and bisexual men in Taiwan who completed a cross-sectional online survey, which included the UCLA Loneliness Scale, Center for Epidemiological Studies Depression Scale (CES-D) and State–Trait Anxiety Inventory (STAI). Confirmatory factor analysis was used to evaluate factorial validity. Convergent validity was examined between the three versions of the UCLA Loneliness Scale and the CES-D and STAI. Known-group validity was investigated with participants’ sexual orientation and educational levels. The unidimensional construct was supported in all three versions of the UCLA Loneliness Scale tested in the present study. Convergent validity was supported as the level of loneliness was correlated with the level of depression and anxiety for all three versions. There were no significant differences between gay and bisexual men, although significant differences were found across different educational levels. The study confirmed that all three versions of the UCLA Loneliness Scale were comparable with satisfactory reliability and validity in Taiwanese sexual-minority men.

## 1. Introduction

Loneliness is a subjective feeling of perceiving discordance between the desired and actual degree of social connectivity [1]. Research has found that loneliness is prevalent among sexual-minority individuals [2,3,4,5,6,7]. Loneliness in sexual-minority individuals may develop in several ways different from that in heterosexual individuals. First, according to minority stress theory [8], sexual-minority individuals may experience victimization and discrimination rooted in heterosexism since childhood [9]. Sexual-minority individuals may also experience ‘microaggressions’ (i.e., more subtle forms of sexual discrimination), which can make targets feel uncomfortable during social interactions [10,11]. Sexual-minority individuals may hide their sexual orientation to avoid experiencing prejudice, which is one pathway to loneliness [8]. Second, the experiences of sexual discrimination may alter the cognitive processes and coping mechanisms of sexual-minority individuals and increase feelings of loneliness. For example, sexual-minority individuals may internalize sexual stigma as their attitude toward themselves, known as “internalized homonegativity”. Bullying victimization may also increase the risk of self-identity confusion [12]. All these cognitive biases may hamper their interaction with others [13]. Third, not only sexual minority stress [2,8] but also intraminority gay community stress [14] may compromise sexual-minority individuals’ mental health and increase feelings of loneliness. Fourth, both sexual minority stress and intraminority gay community stress may impair the functioning of their relationships and result in loneliness directly [2]. Loneliness has been identified as a risk factor for physical health problems [15,16,17], depression and anxiety [16], suicidal ideation [18], sleep disturbance [17], cannabis use [17] and sexual risk behaviors [19,20], among sexual-minority individuals.

According to the social ecological model [21], sexual-minority men’s loneliness is the result of the interactions between individuals and their environments. In Taiwan, sexual orientation bullying [22,23], microaggressions and internalized homonegativity [24] are common. People in Taiwan have also shown their prejudicial attitudes toward sexual-minority individuals during the debate on legalizing same-sex relationships [25,26,27,28,29,30,31]. A previous study in Taiwan found that 60.3% of gay and bisexual young adult men experienced verbal and physical harassment and 34.4% experienced cyber harassment due to their sexual orientation or gender nonconformity during childhood and adolescence [32]. Harassment due to sexual orientation significantly increased the risks of compromised quality of life [32], depression [33], anxiety [34], suicide [34], alcohol [35] and illicit drug use [36], among sexual-minority men in Taiwan. Therefore, it is important to evaluate loneliness among sexual-minority individuals in Taiwan and develop appropriate interventions.

A validated instrument is important to assess the level of loneliness among sexual-minority individuals. A commonly used instrument is the UCLA Loneliness Scale (Version 3) [37]. The use of the UCLA Loneliness Scale (Version 3) has been widely supported by much evidence showing good psychometric properties. For example, the scale has robust psychometric properties, such as scale score reliability, test–retest reliability, convergent validity, and factorial validity, in a Farsi version among healthy participants [38]. Similar psychometric findings have been reported for the Turkish version [39], Japanese version [40], Danish version [41], English version [42,43], Spanish version [44] and many other language versions [42] across different populations (e.g., older people, adolescents and mothers). However, although the UCLA Loneliness Scale (Version 3) has been translated into Chinese, to the best of the present authors’ knowledge, it has never been validated among a Taiwanese population and no language version has ever been psychometrically evaluated with sexual-minority men.

The original UCLA Loneliness Scale (Version 3) is a 20-item version, and this full version has been revised into several short versions [45,46,47]. More specifically, the 8-item and 3-item short-form versions have been proposed as being psychometrically better than the other short versions [45,48]. Additional evidence shows that both the 8-item and 3-item UCLA Loneliness Scale (Version 3), including their Chinese versions, have good psychometric properties [45,46,47,49]. However, the three versions of the UCLA Loneliness Scale (Version 3) have never been compared in any Chinese-speaking population, including Taiwanese sexual-minority men. Therefore, little is known about whether any of the three versions of the UCLA Loneliness Scale (Version 3) are good instruments for assessing loneliness among sexual-minority men in Taiwan. Moreover, it is unclear whether the 8-item and 3-item versions are comparable to the full Chinese language version.

Another psychometric issue for the UCLA Loneliness Scale (Version 3) is whether it has a unidimensional structure. The literature has argued that loneliness should be treated as a unidimensional construct [50]. However, instruments assessing loneliness have been found to be multidimensional [51,52,53], and the UCLA Loneliness Scale (Version 3) is no exception [54,55,56,57]. Scrutinizing the psychometric evidence on the factor structure of the UCLA Loneliness Scale (Version 3), a bifactor structure (i.e., a general factor of loneliness with two method factors of positive wording and negative wording) has been supported (e.g., English version [37], Farsi version [38], Turkish version [39]). Moreover, some evidence shows that the UCLA Loneliness Scale (Version 3) has a unidimensional structure [17,18]. In brief, the full version of the UCLA Loneliness Scale (Version 3) can be viewed as having a unidimensional structure in relation to loneliness, but the unidimensional structure may be or may not be influenced by wording effects. Therefore, additional psychometric evidence on the UCLA Loneliness Scale (Version 3) is needed for sexual-minority men in Taiwan.

The present study examined the psychometric properties of the UCLA Loneliness Scale (Version 3) for its full version (20 items), 8-item version and 3-item version among a sample of gay and bisexual men in Taiwan. More specifically, a unidimensional structure was examined for all three versions. Furthermore, scale score reliability, convergent validity and known-group validity were examined and compared between the three versions of the UCLA Loneliness Scale (Version 3).

## 2. Materials and Methods

### 2.1. Participants and Procedure

The present study recruited participants by posting advertisements on the *Bulletin Board System* (a popular application for sharing messages online in Taiwan), *Facebook*, *LINE* (a popular app for exchanging messages), and the websites of three health promotion centers for sexual-minority individuals from August 2021 to January 2022. The study’s inclusion criterion was being a Taiwanese gay or bisexual man who lived in Taiwan. The exclusion criteria were those who had difficulties in comprehending the purpose of the survey or the content of the present study due to intellectual disability and cognitive dysfunction caused by alcohol and substance use or brain injury. One individual was excluded due to intellectual disability and another was excluded because of alcohol on his breath. In total, 400 gay and bisexual men took part in the present study and provided written informed consent prior to completing the survey. Moreover, all the participants were interviewed by interviewers who received standardized training and were able to encourage the participants to fully complete the survey questionnaires. Therefore, the present data had no missing values. The present sample was generally young (mean age = 30.7 years with SD = 5.94) and consisted of primarily gay men (n = 333; 83.3%). Over half of the participants had completed their undergraduate degree (n = 274; 68.5%) and nearly 15% of the participants had a postgraduate degree (n = 59; 14.8%). The Institutional Review Board of Kaohsiung Medical University Hospital approved the study (KMUHIRB-F(I)-20210119).

### 2.2. Measures

#### 2.2.1. UCLA Loneliness Scale (Version 3)

The UCLA Loneliness Scale (Version 3) contains 20 items (e.g., “I have nobody to talk to”, “I cannot tolerate being so alone”) rated from 1 (never) to 4 (always). Nine items of the UCLA Loneliness Scale (Version 3) are negatively worded and their scores are reverse-coded [37]. A higher score on the UCLA Loneliness Scale (Version 3) indicates higher levels of loneliness. Moreover, prior research indicates that the theoretical framework for the UCLA Loneliness Scale (Version 3) should be unidimensional (i.e., loneliness is a unidimensional construct [50]), although the full version of the UCLA Loneliness Scale (Version 3) has been found to have different structures [54,55,56,57]. The different structures of the full version of the UCLA Loneliness Scale (Version 3) were later found to be confounded by wording effects [37,57,58]. Apart from the full version of the UCLA Loneliness Scale (Version 3), two short versions (i.e., an 8-item version and a 3-item version) have been developed and found to have satisfactory psychometric properties. More specifically, the literature shows that scale score reliability and factorial validity for the UCLA Loneliness Scale (Version 3) were satisfactory in all three versions [37,45,46,47,48,49]. For example, Cronbach’s α was 0.89 to 0.94 (full version) [37], 0.84 (8-item version) [45] and 0.72 to 0.87 (3-item version) [46,47]. Moreover, both the 8-item and 3-item Chinese versions of the UCLA Loneliness Scale (Version 3) have been found to have satisfactory psychometric properties [47,49]. Psychometric properties of the three scales in the present study are reported in the Results section.

The UCLA Loneliness Scale (Version 3) has been translated into Chinese for Taiwanese older people via a standard translation procedure, including forward translation, back translation, expert panel committee, and pilot testing. In the expert panel committee, six experts (including experts in sociology, psychology, and nursing) have discussed whether the translated UCLA Loneliness Scale (Version 3) fits the Taiwan context. Although Taiwanese culture may be different from American culture (where the UCLA Loneliness Scale (Version 3) was developed), prior evidence shows that the translated UCLA Loneliness Scale (Version 3) is applicable and appropriate to assess loneliness among Taiwanese people [59].

#### 2.2.2. Center for Epidemiological Studies Depression Scale (CES-D)

The CES-D contains 20 items rated on a four-point scale (0 = rarely or none of the time (less than 1 day); 4 = most or all of the time (5–7 days)). Responses on the 20 items are summed and a higher score on the CES-D indicates higher levels of depression [60]. The psychometric properties of the CES-D have been shown to be satisfactory [61], including the Chinese version [62,63]. For example, Cronbach’s α was 0.84 [61]. The α was 0.92 (95% confidence interval [CI]: 0.91–0.93) in the present study.

#### 2.2.3. State–Trait Anxiety Inventory State Anxiety (STAI)

The STAI used in the present study contained 20 state anxiety items rated on a four-point scale (1 = almost never; 4 = almost always). Responses on the 20 items are summed and a higher score on the STAI indicates higher levels of state anxiety [64]. The psychometric properties of the STAI have been shown to be satisfactory [65,66], including the Chinese version [67,68,69,70]. For example, Cronbach’s α was 0.95 [66]. The α was 0.95 (95% CI: 0.94–0.96) in the present study.

### 2.3. Data Analysis

Descriptive statistics were firstly applied to summarize the present sample’s characteristics, including their demographics and scores on the measures (including the three versions of the UCLA Loneliness Scale (Version 3), CES-D, and STAI). Descriptive statistics were then applied to examine the score distribution of the 20 items on the UCLA Loneliness Scale. Outlier examination was conducted using the Mean and Standard Deviation (SD) Method: when a participant has his/her UCLA Loneliness Scale total score (either in the full version, 8-item version or 3-item version) over 3 SDs, the participant is identified as an outlier. For psychometric testing, Cronbach’s α and McDonald’s ω were used to determine the scale score reliability of the three versions of the UCLA Loneliness Scale (Version 3). The values of α and McDonald’s ω > 0.7 are acceptable [71]. Confirmatory factor analysis (CFA) with a diagonally weighted least squares estimator was used to examine the unidimensional structure of the three versions of the UCLA Loneliness Scale (Version 3). Factor loadings > 0.3 in the CFA results indicate that the items are necessary in the scale [72,73]. Moreover, several fit indices, including comparative fit index (CFI), Tucker–Lewis index (TLI), root mean square error of approximation (RMSEA) and standardized root mean square residual (SRMR), were used to examine whether the unidimensional structure fit each version of the UCLA Loneliness Scale (Version 3). CFI and TLI > 0.9 with RMSEA and SRMR < 0.08 indicate satisfactory fit and, therefore, support the unidimensional structure [74].

Convergent validity of the three versions of the UCLA Loneliness Scale (Version 3) was carried out using Pearson correlations with the two external criterion measures (i.e., CES-D and STAI). Given that prior research indicates moderate to high correlations between psychological distress and loneliness [37,47], moderate to large effects (i.e., *r* > 0.3) [75] in the Pearson correlations were expected. Lastly, known-group validity of the UCLA Loneliness Scale (Version 3) was tested for the three versions using independent *t*-tests (with the independent variable of sexual orientation) and analyses of variance (with the independent variable of educational level in senior high or below, undergraduate, and postgraduate). Based on the previous literature, it was hypothesized that no differences would be found between gay and bisexual orientation [5] and that individuals with lower levels of educational attainment would report higher levels of loneliness [76]. All the statistical analyses were performed using R software (R Foundation for Statistical Computing, Vienna, Austria). More specifically, CFA was performed using the *lavaan* package [77] and Cronbach’s α and McDonald’s ω were calculated using the *psych* package [78].

## 3. Results

The participants’ scores on the UCLA Loneliness Scale (including full version, 3-item version and 8-item version), CES-D and STAI are reported in Table 1. No outliers were identified in either the 3-item (SD = −1.80 to 2.13) or 8-item version (SD = −2.28 to 2.84); one participant was found to be an outlier in the full version (SD = −2.11 to 3.29).

The item properties of the 20 items in the UCLA Loneliness Scale are reported in Table 2. In summary, the item scores could be viewed as normally distributed (skewness = −0.314 to 0.829; kurtosis = −1.032 to 0.396) with all participants responding to all items on every Likert-type response. Moreover, the mean scores for the items were between 1.65 and 2.75 on the four-point Likert scale. CFA results showed that the 20 items had relatively satisfactory factor loadings in the full version (loading values ranged between 0.40 and 0.72, except for Item 17 (value = 0.28)), the 8-item version (loading values ranged between 0.34 and 0.74), and the 3-item version (loading values ranged between 0.68 and 0.78). Additionally, the unidimensional structure of the UCLA Loneliness Scale (Version 3) was supported for both the full and the 8-item versions, as evidenced by the satisfactory fit indices: CFI = 0.98 and 0.97; TLI = 0.97 and 0.96; RMSEA = 0.060 and 0.070; and SRMR = 0.075 and 0.066. The fit indices for the 3-item version of the UCLA Loneliness Scale (Version 3) were perfectly fit because this model only contains three items, which is a saturated model in the CFA equation (Table 3).

All three versions of the UCLA Loneliness Scale (Version 3) had acceptable scale score reliability (Cronbach’s α and McDonald’s ω (95% CI) = 0.92 and 0.94 (0.93, 0.94) for the full version, respectively; 0.80 and 0.86 (0.84, 0.88) for the 8-item version, respectively; and 0.76 and 0.77 (0.76, 0.86) for the 3-item version, respectively). The convergent validity of the three versions was similar because their correlations with the CES-D score (*r* = 0.66 to 0.68) and those with the STAI score (*r* = 0.47 to 0.51) were all statistically significant (*p*-values < 0.001), with moderate to large effect sizes. The correlations between the three versions of the UCLA Loneliness Scale (Version 3) were all above 0.80 (Table 4). The known-group validity of the UCLA Loneliness Scale (Version 3) also showed similar results across the three versions. All three versions showed that there were no significant differences between gay and bisexual men (*p*-values = 0.22 to 0.89). Moreover, all three versions reported significant differences between participants with a lower educational level (i.e., senior high or below) and those with a higher level (i.e., undergraduate (*p*-values = 0.014 to 0.045) or postgraduate (*p*-values = 0.010 to 0.036)) (Table 5).

## 4. Discussion

The present study demonstrated that the three different versions of the UCLA Loneliness Scale (Version 3) tested have good psychometric properties in terms of factorial validity, convergent validity, scale score reliability, and known-group validity. More specifically, all three versions had a unidimensional factor structure, as supported by the satisfactory fit indices in each CFA (Table 3). In addition, all three versions had similar magnitudes of correlations with depression and anxiety (Table 4) and had similar differences in distinguishing sexual-minority men of lower educational level and higher educational level (Table 5). The three versions of the UCLA Loneliness Scale (Version 3) were also highly associated with each other.

An important issue of using the UCLA Loneliness Scale (Version 3) is to clarify whether it posits a unidimensional construct that can reflect the theoretical concept of loneliness [50]. The present study’s findings indicate that the full version of the UCLA Loneliness Scale (Version 3), together with the two short versions (i.e., 8-item and 3-item versions), share the same unidimensional structure. The findings are consistent with the results of prior studies in Danish, Japanese and Hong Kong samples [40,41,47,49]. More specifically, Arimoto and Tadaka [40] and Lasgaard [41] used CFA and found that the unidimensional structure fit the full version of the UCLA Loneliness Scale (Version 3). Wu and Yao [49] also used CFA and found that the 8-item version of the UCLA Loneliness Scale (Version 3) was unidimensional.

Similar findings were reported for the 3-item version: Liu et al. [47] used CFA and found a perfect fit in the 3-item version of the UCLA Loneliness Scale (Version 3). Although the perfect fit reported by Liu et al. [47] is due to a mathematical issue (i.e., a three-item structure in the CFA is a saturated model and will always fit perfectly with data) [79], the factor loadings reported by Liu et al. [47] and the present study were satisfactory for the 3-item version. Therefore, it can be tentatively concluded that the 3-item version is unidimensional.

Although some previous studies have reported a multidimensional structure for the UCLA Loneliness Scale (Version 3) full version [54,55,56,57], this may be due to wording effects [14]. More specifically, when using CFA to control for wording effects (i.e., there are 9 negatively worded items and 11 positively worded items in the full version), only one general factor representing the trait of loneliness was extracted [37,38,39]. In other words, the full version of the UCLA Loneliness Scale (Version 3) can also be viewed as having a unidimensional loneliness construct.

Apart from the evidence of unidimensionality across the three versions of the UCLA Loneliness Scale (Version 3), the present study extended the literature on the comparable convergent validity and known-group validity for the three versions. Prior evidence has shown that loneliness is moderately to highly associated with psychological distress [37,45,46,47]. Therefore, the moderate to high correlations shown between the three versions of the UCLA Loneliness Scale (Version 3) and the two external criterion instruments (CES-D and STAI) demonstrated the convergent validity of the UCLA Loneliness Scale (Version 3). Moreover, the three versions of the UCLA Loneliness Scale (Version 3) effectively distinguished the different levels of loneliness between sexual-minority men with a higher educational level and those with a lower level. The finding echoes previous evidence in the literature [76]. Furthermore, there were no statistically significant differences in the levels of loneliness between gay and bisexual participants in the present study.

The present study provided psychometric evidence for the full version, 8-item version and 3-item version of the UCLA Loneliness Scale (Version 3). In general, all three versions are useful and can accurately assess the construct of loneliness for sexual-minority men in Taiwan. Although a somewhat lower Cronbach’s α has been reported in the 3-item version, this can be explained by the fact that there are fewer items in the short version. More specifically, Cronbach’s α is highly associated with the number of items in an instrument [80]. Therefore, it is reasonable that the 3-item UCLA Loneliness Scale (Version 3) had poorer scale score reliability than the two longer versions. Nevertheless, the 3-item version still has acceptable scale score reliability (i.e., Cronbach’s α > 0.7) [71]. In other words, the 3-item version of the UCLA Loneliness Scale (Version 3) is still recommended, even with a slightly lower α. Therefore, the consideration of when to use which versions of the UCLA Loneliness Scale (Version 3) will be emphasized based on the time and practical settings for healthcare providers and therapists. More specifically, when time is more restricted and the practical settings are busier, shorter versions of the UCLA Loneliness Scale (Version 3) are recommended. In contrast, if healthcare providers and therapists have sufficient time, using a longer version of the UCLA Loneliness Scale (Version) is better as more information can be obtained.

The present study found that loneliness was significantly associated with depression and anxiety in sexual-minority men. The results confirmed the importance of considering loneliness as a target of the interventions for health in sexual-minority men. Loneliness is a negative result of sexual minority stress and can have prolonged impacts on physical and psychological health. Further efforts to modify the public’s prejudicial attitudes with regard to sexual-minority individuals are warranted. Establishing anti-discrimination laws and policies that protect individuals from sexual-orientation discrimination is of fundamental importance [81]. To help reduce sexuality-related stigma, it is necessary to broaden the public’s understanding of sexual minority culture and awareness of prejudices toward sexual-minority individuals in educational settings, workplaces, and family environments [81,82]. Public health strategies addressing attitudes toward sexual orientation and promoting changes in the public’s attitudes may contribute to diverse affirmative cultures regarding sexual-minority individuals [81,83].

There are some limitations in the present study. First, some important psychometric properties (e.g., test–retest reliability and responsiveness) were not examined. Therefore, future studies are needed to examine whether all three versions of the UCLA Loneliness Scale (Version 3) tested in the present study have satisfactory reproducibility and responsivity. Second, the two external criterion instruments (i.e., CES-D and STAI) were self-administered by the participants. Therefore, the two external criterion instruments have the concerns of single-rater biases, given that the UCLA Loneliness Scale (Version 3) was also completed by the participants themselves. Third, self-identified gay or bisexual men could participate in this study. This study did not inquire about participants’ gender identity. However, research has found that sexual- and gender-minority identities have intersectional impacts on health [84], behaviors [85] and risk of intimate partner violence [86]. Further study is needed to examine the intersectional impacts of sexual- and gender-minority identities on loneliness. Finally, most of the sample was made up of gay men (83.3%) and only a small proportion was bisexual (16.7%). Research has found that bisexual men may be at elevated risk for health disparities and shortage of social resources compared with gay men [87,88]. Although no differences were found in loneliness between the two samples, further study is needed to examine whether the role of loneliness in health differs between gay and bisexual men.

## 5. Conclusions

In conclusion, the three versions of the UCLA Loneliness Scale (Version 3) were found to be satisfactory instruments for assessing loneliness for sexual-minority men in Taiwan. The scale score reliability, convergent validity with psychological distress (depression and anxiety), known-group validity (between educational levels) and unidimensional structure of the three versions (i.e., full version, 8-item version and 3-item version) were fully supported by the present study’s results. Moreover, the present study demonstrated that the three versions share similar psychometric characteristics, except for scale score reliability, and, therefore, were comparable. Healthcare providers and therapists can decide which version of the UCLA Loneliness Scale (Version 3) to use to assess loneliness for sexual-minority men based on time availability and practicality.

## Figures and Tables

**Table 1 ijerph-19-08095-t001:** Participants’ characteristics (N = 400).

Variable	M (SD) or n (%)
Age (year)	30.7 (5.94)
Sexual orientation	
*Gay*	333 (83.3)
*Bisexual*	67 (16.7)
Educational level	
*Senior high or below*	67 (16.8)
*Undergraduate*	274 (68.5)
*Postgraduate*	59 (14.8)
^a^ UCLA Loneliness Scale (Version 3) full version score	2.17 (0.56)
^a^ UCLA Loneliness Scale (Version 3) 8-item version score	2.34 (0.59)
^a^ UCLA Loneliness Scale (Version 3) 3-item version score	2.37 (0.76)
CES-D score	18.30 (11.12)
STAI score	39.19 (12.47)

CES-D = Center for Epidemiological Studies Depression; STAI = State-Trait Anxiety Inventory. ^a^ Loneliness scores were summed and then divided by the total number of scale items.

**Table 2 ijerph-19-08095-t002:** Item properties of the UCLA Loneliness Scale (Version 3) (N = 400).

Item#	M (SD)	n (%)				Skewness	Kurtosis
		Score 1	Score 2	Score 3	Score 4		
Item 1 ^a^	1.65 (0.69)	186 (46.5)	175 (43.8)	34 (8.5)	5 (1.3)	0.829	0.396
Item 2	2.52 (0.95)	67 (16.8)	120 (30.0)	149 (37.3)	64 (16.0)	−0.098	−0.912
Item 3	2.29 (0.94)	95 (23.8)	136 (34.0)	128 (32.0)	41 (10.3)	0.139	−0.929
Item 4	2.60 (0.93)	57 (14.3)	112 (28.0)	165 (41.3)	66 (16.5)	−0.209	−0.786
Item 5 ^a^	1.87 (0.82)	150 (37.5)	167 (41.8)	69 (17.3)	14 (3.5)	0.634	−0.27
Item 6 ^a^	2.32 (0.78)	56 (14.0)	183 (45.8)	139 (34.8)	22 (5.5)	0.08	−0.429
Item 7	1.88 (0.89)	169 (42.3)	130 (32.5)	83 (20.8)	18 (4.5)	0.631	−0.619
Item 8	2.16 (0.93)	109 (27.3)	156 (39.0)	98 (24.5)	37 (9.3)	0.377	−0.738
Item 9 ^a^	1.99 (0.88)	136 (34.0)	156 (39.0)	86 (21.5)	22 (5.5)	0.515	−0.568
Item 10 ^a^	2.07 (0.76)	92 (23.0)	199 (49.8)	99 (24.8)	10 (2.5)	0.234	−0.448
Item 11	2.23 (0.84)	81 (20.3)	173 (43.3)	120 (30.0)	26 (6.5)	0.202	−0.599
Item 12	1.84 (0.85)	165 (41.3)	151 (37.8)	68 (17.0)	16 (4.0)	0.714	−0.281
Item 13	2.36 (0.98)	93 (23.3)	123 (30.8)	132 (33.0)	52 (13.0)	0.073	−1.026
Item 14	2.37 (0.98)	92 (23.0)	123 (30.8)	131 (32.8)	54 (13.5)	0.071	−1.032
Item 15 ^a^	1.89 (0.81)	141 (35.3)	179 (44.8)	65 (16.3)	15 (3.8)	0.642	−0.12
Item 16 ^a^	2.27 (0.91)	86 (21.5)	158 (39.5)	118 (29.5)	38 (9.5)	0.215	−0.754
Item 17	2.75 (0.95)	47 (11.8)	101 (25.3)	158 (39.5)	94 (23.5)	−0.314	−0.798
Item 18	2.67 (0.88)	39 (9.8)	128 (32.0)	161 (40.3)	72 (18.0)	−0.147	−0.698
Item 19 ^a^	1.88 (0.84)	153 (38.3)	158 (39.5)	74 (18.5)	15 (3.8)	0.618	−0.393
Item 20 ^a^	1.93 (0.87)	150 (37.5)	144 (36.0)	90 (22.5)	16 (4.0)	0.503	−0.696

^a^ These item scores have been reverse-coded.

**Table 3 ijerph-19-08095-t003:** Confirmatory factor analysis results of the UCLA Loneliness Scale (Version 3) and its short versions (N = 400).

	Full Version	8-Item Version	3-Item Version
**Factor loading**			
Item 1	0.49	--	--
Item 2	0.70	0.74	0.68
Item 3	0.65	0.63	--
Item 4	0.67	--	--
Item 5	0.64	--	--
Item 6	0.40	--	--
Item 7	0.69	--	--
Item 8	0.69	--	--
Item 9	0.45	0.46	--
Item 10	0.58	--	--
Item 11	0.68	0.71	0.78
Item 12	0.59	--	--
Item 13	0.70	--	--
Item 14	0.69	0.69	0.71
Item 15	0.69	0.61	--
Item 16	0.66	--	--
Item 17	0.28	0.34	--
Item 18	0.47	0.47	--
Item 19	0.70	--	--
Item 20	0.72	--	--
**Fit statistics**			
χ^2^ (df)	416.46 (170)	59.47 (20)	0 (0) ^a^
*p*-value	<0.001	<0.001	-- ^a^
CFI	0.98	0.97	1.00 ^a^
TLI	0.97	0.96	1.00 ^a^
RMSEA	0.060	0.070	0.000 ^a^
90% CI of RMSEA	0.053, 0.068	0.050, 0.091	0.000, 0.000 ^a^
SRMR	0.075	0.066	0.000 ^a^

^a^ Perfect fit statistics because this model only contains three items, which is a saturated model in the equation of confirmatory factor analysis. CFI = comparative fit index; TLI = Tucker–Lewis index; RMSEA = root mean square error of approximation; SRMR = standardized root mean square residual.

**Table 4 ijerph-19-08095-t004:** Convergent validity of the three versions of the UCLA Loneliness Scale (Version 3) (N = 400).

	Full Version (α = 0.92; ω = 0.94)	8-Item Version (α = 0.80; ω = 0.86)	3-Item Version (α = 0.76; ω = 0.77)	CES-D (α = 0.92)	STAI (α = 0.95)
Full version	--				
8-item version	0.94	--			
3-item version	0.86	0.89	--		
CES-D	0.68	0.66	0.66	--	
STAI	0.51	0.48	0.47	0.70	--

CES-D = Center for Epidemiological Studies Depression; STAI = State-Trait Anxiety Inventory. All *p*-values < 0.001.

**Table 5 ijerph-19-08095-t005:** Comparing the three versions of the UCLA Loneliness Scale (Version 3) between sexual orientation and educational level (N = 400).

	Full Version		8-Item Version		3-Item Version	
	M (SD)	*t*/F (*p*-Value)	M (SD)	*t*/F (*p*-Value)	M (SD)	t/F (*p*-Value)
**Sexual orientation**		0.92 (0.36)		0.14 (0.89)		1.23 (0.22)
Gay (n = 333)	2.16 (0.55)		2.33 (0.58)		2.35 (0.75)	
Bisexual (n = 67)	2.23 (0.58)		2.35 (0.63)		2.48 (0.82)	
**Educational level ^a^**		5.09 (0.007)		3.83 (0.02)		4.70 (0.01)
Senior high or below (n = 67)	2.36 (0.52)		2.51 (0.57)		2.61 (0.73)	
Undergraduate (n = 274)	2.15 (0.56)		2.31 (0.58)		2.35 (0.77)	
Postgraduate (n = 59)	2.08 (0.55)		2.25 (0.58)		2.21 (0.73)	

^a^ Bonferroni adjustment tests for full version: senior high or below was significantly higher than undergraduate (*p* = 0.014) and postgraduate (*p* = 0.012); for 8-item version: senior high or below was significantly higher than undergraduate (*p* = 0.045) and postgraduate (*p* = 0.036); for 3-item version: senior high or below was significantly higher than undergraduate (*p* = 0.041) and postgraduate (*p* = 0.010).

## Data Availability

The data will be available upon reasonable request to the corresponding authors.

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
