# Peer review of "Psychometric Evaluation of Three Versions of the UCLA Loneliness Scale (Full, Eight-Item, and Three-Item Versions) among Sexual Minority Men in Taiwan"

_ijerph, 2022, doi:10.3390/ijerph19138095_

Round 1

Reviewer 1 Report

I consider the study to be interesting, but there are theoretical aspects that do not fully justify its implementation.
For example, I consider the introduction to be incomplete. I think it should justify why homosexual and bisexual men may feel a higher level of loneliness. Point out the characteristics of homosexuality and bisexuality. It is important to clarify that there are many aspects such as discrimination or homophobic bullying, internalised homophobia, which explain the high levels of loneliness, depression and anxiety in gay and bisexual men.
In addition, it should be indicated whether there are studies showing that bisexual men show the same levels of loneliness as gay men.
Therefore, more information would be needed in the introduction to better justify why the sample of sexual minority men has been chosen.
Regarding the methodology
I think it would be important to indicate the average age of the participants in the methodology section. The authors include this information in the results, however, I understand that the socio-demographic data should be included in the description of the sample. 
With respect to the discussion, I consider that it is well presented, although it would be interesting if it could be expanded as more information is provided in the introduction.

Author Response

We appreciated your valuable comments. As discussed below, we have revised our manuscript with underlines based on your suggestions. Please let us know if we need to provide anything else regarding this revision.

Comment 1

I consider the study to be interesting, but there are theoretical aspects that do not fully justify its implementation. For example, I consider the introduction to be incomplete. I think it should justify why homosexual and bisexual men may feel a higher level of loneliness. Point out the characteristics of homosexuality and bisexuality. It is important to clarify that there are many aspects such as discrimination or homophobic bullying, internalised homophobia, which explain the high levels of loneliness, depression and anxiety in gay and bisexual men.
Response

Thank you for your comment. We revised Introduction section and added the contents as below explain the high levels of loneliness, depression and anxiety in gay and bisexual men. Please refer to 60-89.

“Loneliness in sexual minority individuals may develop in several ways different from that in heterosexual individuals. First, according to minority stress theory [8], sexual minority individuals may experience victimization and discrimination rooted in heterosexism since childhood [9]; moreover, sexual discrimination may exist in a lower intensity and more subtle nature, known as “microaggressions” to make sexual minority individuals feel uncomfortable in social contacts with others [10,11]. That may hide their sexual orientation in order to avoid prejudice events [8] and loneliness may develop consequently. Second, the experiences of sexual discrimination may alter the cognitive processes and coping mechanisms of sexual minority individuals and increase lonely feeling. For example, sexual minority individuals may internalize sexual stigma as their attitude toward themselves, known as “internalized homonegativity.” Bullying victimization may also increase the risk of self-identity confusion [12]. All these cognitive biases may hamper their interaction with others [13]. Third, not only sexual minority stress [2,8] but also intraminority gay community stress [14] may compromise sexual minority individuals’ mental health and increase the feeling of loneliness. Fourth, both sexual minority stress and intraminority gay community stress may impair the functioning of their relationships and result in loneliness directly [2]. Loneliness has been identified as a risk factor for physical health problems [15-17], depression and anxiety [16], suicidal ideation [18], sleep disturbance [17], cannabis use [17], and sexual risk behaviors [19,20], among sexual minority individuals.

According to social ecological model [21], sexual minority men’s loneliness is the result of the interactions between individuals and their environments. In Taiwan, sexual orientation bullying [22,23] and microaggression and internalized homonegativity [24] are common. People in Taiwan have also shown their discriminant attitudes toward sexual minority individuals during the debate on legalizing the same-sex relationships [25-31]. Living in such an unfriendly environment, a high proportion of sexual minority men in Taiwan suffer from compromised quality of life [32], depression [33], anxiety [34], suicide [34], alcohol [35] and illicit drug use [36]. Therefore, it is important to evaluate loneliness among sexual minority individuals in Taiwan and assist them in improving loneliness.”

Comment 2

It should be indicated whether there are studies showing that bisexual men show the same levels of loneliness as gay men.
Response

Thank you for your comment. The present study did not find the significant difference between gay and bisexual men; however, previous studies have found that bisexual men may be at elevated risk for health disparities and shortage of social resources compared with gay men (described below). One of the possible reasons accounting for the discrepancy between this and previous studies is that most of the sample is gay (n=333) men and only a small proportion is bisexual (n=67). We addressed this non-balanced distribution in the sample as a limitation. Please refer to line 377-381.

Last, most of the sample is gay men (83.3%) and only a small proportion is bisexual (16.7%). Research has found that bisexual men may be at elevated risk for health disparities and shortage of social resources compared with gay men [88,89]. Although no differences were found in loneliness between the two samples, further study is needed to examine whether the role of loneliness in health differs between gay and bisexual men.”

Comment 3

Regarding the methodology
I think it would be important to indicate the average age of the participants in the methodology section. The authors include this information in the results, however, I understand that the socio-demographic data should be included in the description of the sample. 
Response

Thank you for your comment. We moved the socio-demographic data of the sample from Results section to 2.1. Participants and Procedure. Please refer to line 151-154.

Comment 4

With respect to the discussion, I consider that it is well presented, although it would be interesting if it could be expanded as more information is provided in the introduction.

Response

Thank you for your comment. We expanded the Discussion section and added the contents as below. Please refer to line 351-363.

“The present study found that loneliness was significantly associated with depression and anxiety in sexual minority men. The result confirmed the importance of considering loneliness as a target of the interventions for health in sexual minority men. Loneliness is one of negative results of sexual minority stress and can result in prolonged impacts on physical and psychological health. Further efforts to modify the public’s discriminant attitudes with regard to sexual minority individuals are warranted. Establishing anti-discrimination laws and policies that protect the individuals from sexual orientation discrimination is of fundamental importance [82]. To help reduce sexuality-related stigma, it is necessary to broaden the public’s understanding of sexual minority culture and awareness of prejudices toward sexual minority individuals in educational settings, workplaces, and family environments [82,83]. Public health strategies addressing attitudes to sexual orientation and promoting the changes of attitudes in the public may contribute to diverse affirmative cultures regarding sexual minority individuals [82,84].”

Reviewer 2 Report

This is a great paper but it is far more focused on methods than the population. I'm not sure why you have chosen to focus on gay and bisexual men when you only spend three sentences in the beginning of the paper describing health disparities of this population. Either focus on the population all sections or do not mention the population at all. Otherwise it is adding to overall negative stereotypes of the population.

Author Response

We appreciated your valuable comments. As discussed below, we have revised our manuscript with underlines based on your suggestions. Please let us know if we need to provide anything else regarding this revision.

Comment

This is a great paper but it is far more focused on methods than the population. I'm not sure why you have chosen to focus on gay and bisexual men when you only spend three sentences in the beginning of the paper describing health disparities of this population. Either focus on the population all sections or do not mention the population at all. Otherwise it is adding to overall negative stereotypes of the population.

Response

Thank you for your comment. We have added the introduction regarding loneliness in sexual minority individuals and the necessity of examining loneliness in this population as below. Please refer to 60-89.

“Loneliness in sexual minority individuals may develop in several ways different from that in heterosexual individuals. First, according to minority stress theory [8], sexual minority individuals may experience victimization and discrimination rooted in heterosexism since childhood [9]; moreover, sexual discrimination may exist in a lower intensity and more subtle nature, known as “microaggressions” to make sexual minority individuals feel uncomfortable in social contacts with others [10,11]. That may hide their sexual orientation in order to avoid prejudice events [8] and loneliness may develop consequently. Second, the experiences of sexual discrimination may alter the cognitive processes and coping mechanisms of sexual minority individuals and increase lonely feeling. For example, sexual minority individuals may internalize sexual stigma as their attitude toward themselves, known as “internalized homonegativity.” Bullying victimization may also increase the risk of self-identity confusion [12]. All these cognitive biases may hamper their interaction with others [13]. Third, not only sexual minority stress [2,8] but also intraminority gay community stress [14] may compromise sexual minority individuals’ mental health and increase the feeling of loneliness. Fourth, both sexual minority stress and intraminority gay community stress may impair the functioning of their relationships and result in loneliness directly [2]. Loneliness has been identified as a risk factor for physical health problems [15-17], depression and anxiety [16], suicidal ideation [18], sleep disturbance [17], cannabis use [17], and sexual risk behaviors [19,20], among sexual minority individuals.

According to social ecological model [21], sexual minority men’s loneliness is the result of the interactions between individuals and their environments. In Taiwan, sexual orientation bullying [22,23] and microaggression and internalized homonegativity [24] are common. People in Taiwan have also shown their discriminant attitudes toward sexual minority individuals during the debate on legalizing the same-sex relationships [25-31]. Living in such an unfriendly environment, a high proportion of sexual minority men in Taiwan suffer from compromised quality of life [32], depression [33], anxiety [34], suicide [34], alcohol [35] and illicit drug use [36]. Therefore, it is important to evaluate loneliness among sexual minority individuals in Taiwan and assist them in improving loneliness.”

Reviewer 3 Report

Manuscript ID: IJERPH-1644291

The study presents the validation of the UCLA Loneliness Scale in its three versions, the full 20-item version, and the two short 8-item and 3-item versions, in gay and bisexual men in Taiwan. The adaptation of psychological questionnaires should take into account the relevant features of the population they are addressed to; sexual orientation being an important characteristic when studying the loneliness variable.

The work is a neat, well-written and well-founded piece of research.

I have just three minor suggestions which might help improve the clarity and projection of the study.

  1. When describing the instruments, the potential number of answers used in the Likert-scales is even, when usually, what characterises Likert-scales is that the number of options is odd. I suggest that authors just delete the words “Likert Scale” in the description of the instruments.
  2. Most of the sample is gay (n=333) men and only a small proportion is bisexual (n=67). This non-balanced distribution in the sample should be included as a limitation, even though no differences were found in loneliness between the two samples.
  3. Finally, it would be useful to include the McDonald's omega reliability coefficients of the questionnaires.

Overall, I think this work will make a valuable and insightful contribution to a more-than-relevant field of research.

Author Response

We appreciated your valuable comments. As discussed below, we have revised our manuscript with underlines based on your suggestions. Please let us know if we need to provide anything else regarding this revision.

Comment 1

When describing the instruments, the potential number of answers used in the Likert-scales is even, when usually, what characterises Likert-scales is that the number of options is odd. I suggest that authors just delete the words “Likert Scale” in the description of the instruments.

Response

Thank you for your comment. We have deleted “Likert Scale.” Please refer to line 158-159, line 187, and line 195.

Comment 2

Most of the sample is gay (n=333) men and only a small proportion is bisexual (n=67). This non-balanced distribution in the sample should be included as a limitation, even though no differences were found in loneliness between the two samples.

Response

Thank you for your comment. We have added it as the limitation of this study as below. Please refer to line 377-381.

Last, most of the sample is gay men (83.3%) and only a small proportion is bisexual (16.7%). Research has found that bisexual men may be at elevated risk for health disparities and shortage of social resources compared with gay men [88,89]. Although no differences were found in loneliness between the two samples, further study is needed to examine whether the role of loneliness in health differs between gay and bisexual men.”

Comment 3

Finally, it would be useful to include the McDonald's omega reliability coefficients of the questionnaires.

Response

We have now provided the information.

“For psychometric testing, internal consistency via Cronbach’s α and McDonald’s ω was used to examine the internal consistency of the three versions of UCLA Loneliness Scale (Version 3).”

  1. “For psychometric testing, internal consistency via Cronbach’s α and McDonald’s ω was used to examine the internal consistency of the three versions of UCLA Loneliness Scale (Version 3).” Please refer to line 208-209.
  2. “Cronbach’s α with McDonald’s ω were calculated using the psych package” Please refer to line 232-233.
  3. “All three versions of the UCLA Loneliness Scale (Version 3) had acceptable internal consistency (Cronbach’s α and McDonald’s ω [95% CI] = 0.92 and 0.94 [0.93, 0.94] for the full version, respectively; 0.80 and 0.86 [0.84, 0.88] for the 8-item version, respectively; and 0.76 and 0.77 [0.76, 0.86] for the 3-item version, respectively).” Please refer to line 263-266.

Reviewer 4 Report

The authors conducted a study of the UCLA Loneliness Scale in a sample of sexual minority adults in Taiwan. I recommend the following revisions to strengthen the paper:

1. Overall, the introduction needs to make a stronger case for the relevance of this measure to sexual minority men.

2. Related to the pervious comment, more background on stigma and minority stress is recommended.

3. More clarity on the exclusion criteria ("those who had difficulties in comprehending the purpose or the survey content of the present study due to intellectual disability and cognitive dysfunction caused by alcohol and substance use or brain injury") would be very helpful. Were exclusions for these reasons common?

4. One or two examples of items from the loneliness scale would be helpful.

5. How was missingness handled? Was there any imputation used?

6. How were outliers assessed?

7. Overall, while the discussion describes the analytic/methodological aspects of the study well, there needs to me much more discussion on the relevance of loneliness to sexual minority adults in Taiwan, and drivers of loneliness in this population (such as ostracization, stigma, etc.).

8. More discussion on policy recommendations would also strengthen the paper.

Author Response

We appreciated your valuable comments. As discussed below, we have revised our manuscript with underlines based on your suggestions. Please let us know if we need to provide anything else regarding this revision.

Comment

  1. Overall, the introduction needs to make a stronger case for the relevance of this measure to sexual minority men.
  2. Related to the pervious comment, more background on stigma and minority stress is recommended.

Response

Thank you for your comment. We have added the introduction regarding loneliness in sexual minority individuals and the necessity of examining loneliness in this population as below. Please refer to 60-89.

“Loneliness in sexual minority individuals may develop in several ways different from that in heterosexual individuals. First, according to minority stress theory [8], sexual minority individuals may experience victimization and discrimination rooted in heterosexism since childhood [9]; moreover, sexual discrimination may exist in a lower intensity and more subtle nature, known as “microaggressions” to make sexual minority individuals feel uncomfortable in social contacts with others [10,11]. That may hide their sexual orientation in order to avoid prejudice events [8] and loneliness may develop consequently. Second, the experiences of sexual discrimination may alter the cognitive processes and coping mechanisms of sexual minority individuals and increase lonely feeling. For example, sexual minority individuals may internalize sexual stigma as their attitude toward themselves, known as “internalized homonegativity.” Bullying victimization may also increase the risk of self-identity confusion [12]. All these cognitive biases may hamper their interaction with others [13]. Third, not only sexual minority stress [2,8] but also intraminority gay community stress [14] may compromise sexual minority individuals’ mental health and increase the feeling of loneliness. Fourth, both sexual minority stress and intraminority gay community stress may impair the functioning of their relationships and result in loneliness directly [2]. Loneliness has been identified as a risk factor for physical health problems [15-17], depression and anxiety [16], suicidal ideation [18], sleep disturbance [17], cannabis use [17], and sexual risk behaviors [19,20], among sexual minority individuals.

According to social ecological model [21], sexual minority men’s loneliness is the result of the interactions between individuals and their environments. In Taiwan, sexual orientation bullying [22,23] and microaggression and internalized homonegativity [24] are common. People in Taiwan have also shown their discriminant attitudes toward sexual minority individuals during the debate on legalizing the same-sex relationships [25-31]. Living in such an unfriendly environment, a high proportion of sexual minority men in Taiwan suffer from compromised quality of life [32], depression [33], anxiety [34], suicide [34], alcohol [35] and illicit drug use [36]. Therefore, it is important to evaluate loneliness among sexual minority individuals in Taiwan and assist them in improving loneliness.”

Comment

  1. More clarity on the exclusion criteria ("those who had difficulties in comprehending the purpose or the survey content of the present study due to intellectual disability and cognitive dysfunction caused by alcohol and substance use or brain injury") would be very helpful. Were exclusions for these reasons common?

Response

Thank you for your comment. There were two individuals who were excluded from this study. We have added the introduction as below. Please refer to line 145-146.

One individual was excluded due to low mentality and another was excluded because of alcohol on his breath.”

Comment

  1. One or two examples of items from the loneliness scale would be helpful.

Response

Thank you for your comment. We have added two examples of items as below. Please refer to line 158.

(e.g., “I have nobody to talk to” “I cannot tolerate being so alone”)

Comment

  1. How was missingness handled? Was there any imputation used?

Response

Because the present study was conducted using rigorous interviews (i.e., the interviewers were all well trained and had the ability to encourage the participants to complete the survey questions without any missing), we did not handle any missing data in the present study. Accordingly, we did not impute any data. However, we understand that it is important for readers to know this information. Therefore, we have now clearly stated that the present data had no missing in the revised manuscript. Please refer to line 148-151.

Moreover, all the participants were interviewed by interviewers who received standardized training and were able to encourage the participants to fully complete the survey questionnaires. Therefore, the present data had no missing values.

Comment

  1. How were outliers assessed?

Response

We used the Mean and Standard Deviation Method to assess outliers for the three versions of UCLA Loneliness Scale. When a participant has UCLA Loneliness Scale total score (either in the full version, 8-item version, or 3-item version) over 3 standard deviations (SDs), the participant is identified as an outlier. According to this method, only one participant was identified as an outlier in the full version of UCLA Loneliness Scale (with 3.28 SDs). We have now reported the information of outliers in the revised manuscript.

“Outlier examination was conducted using Mean and Standard Deviation (SD) Method: When a participant has his/her UCLA Loneliness Scale total score (either in the full version, 8-item version, or 3-item version) over 3 SDs, the participant is identified as an outlier.” Please refer to line 205-208.

“Moreover, no outliers were identified in either 3-item (SD = -1.80 to 2.13) or 8-item version (SD = -2.28 to 2.84); one participant was found to be an outlier in the full version (SD = -2.11 to 3.29).” Please refer to line 236-238.

Comment

  1. Overall, while the discussion describes the analytic/methodological aspects of the study well, there needs to me much more discussion on the relevance of loneliness to sexual minority adults in Taiwan, and drivers of loneliness in this population (such as ostracization, stigma, etc.).

Response

Thank you for your comment. We have added the contents regarding loneliness among sexual minority adults in Taiwan as below.

According to social ecological model [21], sexual minority men’s loneliness is the result of the interactions between individuals and their environments. In Taiwan, sexual orientation bullying [22,23] and microaggression and internalized homonegativity [24] are common. People in Taiwan have also shown their discriminant attitudes toward sexual minority individuals during the debate on legalizing the same-sex relationships [25-31]. Living in such an unfriendly environment, a high proportion of sexual minority men in Taiwan suffer from compromised quality of life [32], depression [33], anxiety [34], suicide [34], alcohol [35] and illicit drug use [36]. Therefore, it is important to evaluate loneliness among sexual minority individuals in Taiwan and assist them in improving loneliness.Please refer to line 81-89.

The present study found that loneliness was significantly associated with depression and anxiety in sexual minority men. The result confirmed the importance of considering loneliness as a target of the interventions for health in sexual minority men. Loneliness is one of negative results of sexual minority stress and can result in prolonged impacts on physical and psychological health. Further efforts to modify the public’s discriminant attitudes with regard to sexual minority individuals are warranted. Please refer to line 351-356.

Comment

  1. More discussion on policy recommendations would also strengthen the paper.

Response

Thank you for your comment. We have added more discussion on policy recommendations as below. Please refer to line 356-363.

Establishing anti-discrimination laws and policies that protect the individuals from sexual orientation discrimination is of fundamental importance [82]. To help reduce sexuality-related stigma, it is necessary to broaden the public’s understanding of sexual minority culture and awareness of prejudices toward sexual minority individuals in educational settings, workplaces, and family environments [82,83]. Public health strategies addressing attitudes to sexual orientation and promoting the changes of attitudes in the public may contribute to diverse affirmative cultures regarding sexual minority individuals [82,84].”

Round 2

Reviewer 1 Report

This paper has been improved.

Author Response

Thank you for your comments.

Reviewer 4 Report

The authors have addressed my concerns.

Author Response

Thank you for your comments.